# OpenReview forum: "Token Highlighter: Inspecting and Mitigating Jailbreak Prompts for Large Language Models"
_NeurIPS.cc/2024/Workshop/SafeGenAi — SafeGenAi Poster_

### Official Review · Reviewer_apZ7 · 2024-10-09
**Comprehensive paper.**

**Rating:** 6
**Confidence:** 3

**Review:**

This work proposes a new defense method called "Token highlighter" which works by reducing the importance of tokens that tend to produce an affirmative answer from the LLM. Through extensive evaluation by multiple attacks and comparison with defenses the authors show that the token highlighter approach achieves maximum alpaca win rate while maintaining least ASR.

The method are results are interesting and useful to the community. Only possible limitation can be the assumption that "successful jailbreaks succeed by tricking LLM to produce an affirmative answer" might not hold true for all datasets and scenarios, thus asking for more extensive evaluation other than alpaca-win-rate alone.

---

### Official Review · Reviewer_3oWX · 2024-10-09
**Review for Paper 82**

**Rating:** 6
**Confidence:** 4

**Review:**

**Summary**

The paper proposes an algorithm that identifies jailbreak prompts and mitigates their attack effect.

-----
**Strength**

(1) Under the assumption that the attackers try to make the model respond with an affirmative response, the algorithm shows a promising tradeoff between accuracy and robustness.

(2) The writing is clear and the proposed algorithm is easy to follow.

----
**Weakness**

(1) There are more advanced attacks that work beyond affirmative beginnings, such as those trick models into role-play. I wonder how this algorithm performs when faced with more adaptive attacks.

(2) Backpropagation through the whole model is still too expensive for large models and frequent queries, The algorithm is hard to scale.

---

### Official Review · Reviewer_hQHo · 2024-10-10
**Recommend accepting, a strong jailbreak defense, however, a more reliable evaluation and analysis would improve the work**

**Rating:** 7
**Confidence:** 4

**Review:**

This work proposes a method to defend against LLM jailbreaks. The authors showed that it leads to a favorable attack success rate reduction compared to other approaches.

- The method is simple, conceptually solid, inexpensive, and performs well on AlpacaEval vs ASR plot. It is the main strength of the paper.
- The work is written clearly.

Areas for improvement:
- The method is only evaluated on 2 models: llama 2 and another model derived from llama 2. A more diverse set of models would make the results a lot more robust. Especially consider evaluating on models with a different pre-training pipeline. An analysis of the relationship between relative performance of the proposed method vs other defense methods and model scale would be interesting to see though not a necessity.
- It would be good to have an analysis on which model affirmations work vs not work, how many tokens do you need in the affirmation, how sensitive is the method to the affirmation selection.
- An analysis of how exactly the model deteriorates with this method would be interesting to see. For example, sort benign requests by top reduction in preference over baseline as evaluated by the judge, and show top requests and responses.
- I'm not sure if AlpacaEval with MT-Bench is the top method to measure LLM deterioration anymore. It might be the case that it's still the best or one of the best options, but I'd encourage authors to look around for more sensitive methods. For example, consider measuring model performance in domains that require reasoning, such as science/math. Maybe multiple-choice question answering eval could be one option.